# Variable Length and Variable Quality Audio Steganography

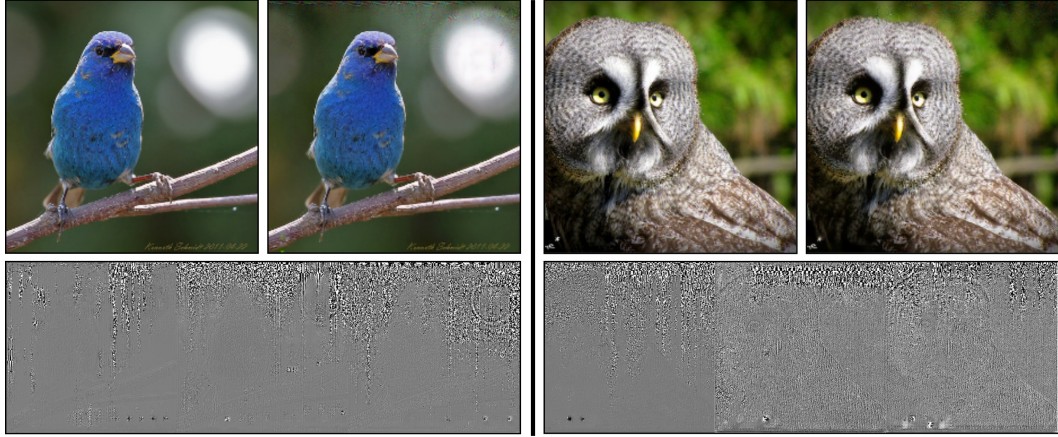

Figure 1: Hiding and uncovering audio data inside an image via VLVQ. Left: Original image. Right: Image with the audio data hidden inside. Bottom: 19 seconds of hidden audio extracted from the image on the right.

## Abstract

Steganography is the task of hiding and recovering secret data inside a non-secret container data while making imperceptible changes to the container. When using steganography to hide audio inside an image, current approaches neither allow the encoding of a signal with variable length nor allow making a trade-off between secret data reconstruction quality and imperceptibility in the changes made to the container image. To address this problem, we propose VLVQ (Variable Length Variable Quality Audio Steganography), a deep learning based steganographic framework capable of hiding variable-length audio inside an image by training the network to iteratively encode and decode the audio data from the container image. Complementary to the standard reconstruction loss, we propose an optional conditional loss term that allows the users to make quality trade-offs between audio and image reconstruction on inference time, without needing to train a separate model for each trade-off setups. Our experiments on ImageNet and AudioSet demonstrate VLVQ's ability to retain reasonable image quality (28.99 $psnr$) and audio reconstruction quality (23.79 $snrseg$) while encoding 19 seconds of audio. We also show VLVQ's capability to generalize to signals longer than what is seen during training.

## 1 Introduction

Natural images are redundant (Ruderman, 1994). They contain predictable structures, repeating textures, and non-uniform color distribution that contribute to their low overall entropy. Meanwhile, the human visual system is insensitive to minor changes to its visual field. These features make steganography, the science of hiding a secret data within a non-secret container data, possible. Several researches have hidden various data inside images such as binary messages (Zhu et al., 2018),

separate images (Baluja, 2017; Tamimi et al., 2013; Wu et al., 2018; Zhang et al., 2019a;b; Duan et al., 2019), and audio signals (Kaul & Bajaj, 2013; Santosa & Bao, 2005; Huu et al., 2019). Here, we focus on the task of hiding audio inside images.

Many algorithms have successfully hidden audio data inside images, especially with the recent advances on deep learning based steganography. However, the use of deep learning restricts these algorithms to only receiving input dimensions that are allowed by the design of the network architecture, which usually have fixed dimensions. This condition makes these methods difficult to operate in the real world, where the signals may easily vary in length. Naively applying these traditional models iteratively on different chunks of the audio fails to work, as the input distribution starts to diverge from that of the training set.

Furthermore, as the bottleneck of most steganographic systems comes from the container data's capacity, hiding a varying length signal inside a fixed-sized container requires the capability to trade-off audio quality and image fidelity. One approach to do so is by changing the coefficients of the different loss terms that each control the reconstruction quality of the container and the secret data. However, this requires the training of several systems with different coefficients, which is not only time and memory consuming, but also fails to cover the full continuous range of possible coefficient combinations.

We propose a new steganographic framework VLVQ (Variable Length Variable Quality Audio Steganography) to encode variable length audio signals in an image and make audio quality and image fidelity trade-off with a single model on inference time. Figure 1 demonstrates an example of VLVQ encoding 19 seconds of audio inside an 224x224 image then decoding it back.

Given the initial container image, VLVQ iteratively embeds an audio chunk inside the image until all of the audio has been encoded. To recover the audio back from this final container image, VLVQ isolates the latest audio chunk embedded inside the container, and update the container image to feed itself back into the decoder recursively. This is repeated until all audio is recovered. An important design choice is to use the trade-off coefficient that linearly weighs the image and audio reconstruction quality as an input to the network. By randomly sampling this coefficient at each training step and feeding it as input to the model, a single model can learn a distribution of solutions.

Section 2 reviews previous works related to VLVQ. Section 3 describes the problem setup and VLVQ's methodology in depth. Then, Section 4 evaluates our approach on ImageNet (Russakovsky et al., 2015) and AudioSet (Gemmeke et al., 2017), demonstrating reasonable reconstruction quality even on signals longer than what is seen during training. Finally, Section 5 discusses various observations and possible directions for future work.

## 2   RELATED WORKS

**Image Steganography.** Steganography is the task of encoding a secret data inside a non-secret container data without making noticeable changes to the container. Image steganography describes a scenario that uses an image as the container that carries the secret data. The limitations of the human visual system in noticing slight changes in brightness or color among many pixels make these image steganography methods effective. Traditional image steganography methods exploit the image spatial domain by modifying the least significant bits (LSB) of the image, or the frequency domain with Discrete Fourier Transform (DFT), Discrete Cosine Transform (DCT), and Discrete Wavelet Transform (DWT). With the help of recent advancements in learning-based algorithms, steganographic systems have seen tremendous advancements. Many of the learning-based steganographic systems employs deep neural networks that are trained end-to-end with reconstruction losses that minimize the change in the container while maximizing the similarity between the recovered secret data and the ground truth. Baluja (2017) uses convolutional architecture to hide images inside images, while Zhu et al. (2018) adopts differentiable augmentations to hide and recover binary data inside images with robustness to image corruptions. Different approaches make improvements in architectural designs (Duan et al., 2019; Cui et al., 2020; Wu et al., 2018) while some adopt Generative Adversarial Networks (GANs) (Goodfellow et al., 2014) to make the changes further imperceptible.

**Hiding Audio in Image.** Hiding audio signal inside images is a specific instance of image steganography. Similar to image steganography in general, methods such as modifying the least significant bits (LSB) (Kaul & Bajaj, 2013) as well as wavelet transformations (Kaul & Bajaj, 2013; Santosa &

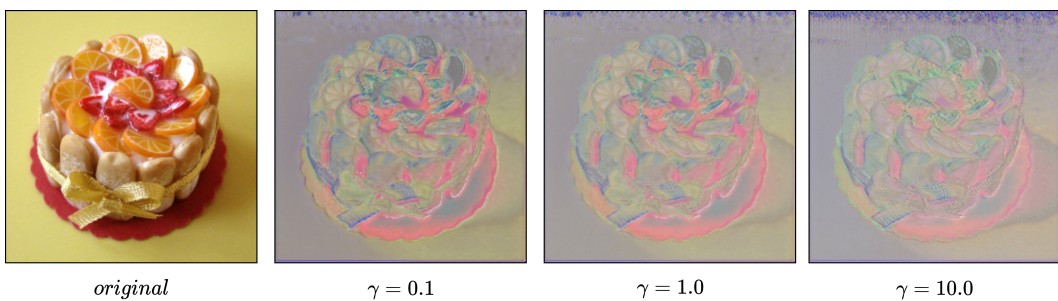



*original*      $\gamma = 0.1$      $\gamma = 1.0$      $\gamma = 10.0$



Figure 2: Difference between the original image and the final container image containing 19 seconds of audio, inferenced using the same model with different conditions. The additional artifact encoding for the STFT features (top part of the difference) becomes more apparent with larger values of $\gamma$, a coefficient used to trade-off between audio quality and image fidelity. (Left: $\gamma = 0.1$. Center $\gamma = 1.0$. Right: $\gamma = 10.0$)

Bao, 2005) have been applied to the task. Deep learning based approaches such as Huu et al. (2019) encodes STFT features of audio into the container image with a convolutional architecture. Ye et al. (2019) incorporates GANs. Still, these methods are limited to fixed dimensional inputs, prohibiting its applications in the real world with audio signals that vary in length.

**Loss-Conditional Training.** Generally, deep learning handles multi-objective training by linearly combining the different sub-objective terms with their respective coefficients. If the coefficient values are closely tied to the model's use cases, one must train multiple models with different coefficient combinations to cover all such cases, which is computationally expensive and impractical. Loss-conditional training resolves this by conditioning the model on the randomly sampled coefficients during training. Babaeizadeh & Ghiasi (2018) first adopted loss-conditional training in style transfer to make the style and content coefficients in style transfer adjustable at inference time. Dosovitskiy & Djolonga (2019) then proposed a general framework for loss-conditional training, demonstrating its applications in autoencoder, image compression, and style transfer. Lee et al. (2020) adopts loss-conditional training in creating a dynamic residual network with adaptable sparsity. Since the importance of audio data reconstruction quality varies among the different use cases of steganography (e.g., in speaker identification, high audio quality may not be necessary as long as the voice is identifiable), we adopt an optional loss-conditional training process to allow the trade-off between these two objectives at inference time.

## 3 METHOD

In this section, we describe the VLVQ (Variable Length and Variable Quality Audio Steganography) framework in detail. We first describe the problem setting, then elaborate on the specific methods. Please refer to Algorithm 1 for an overview of the methods.

### 3.1 PROBLEM SETUP

In steganography, a system $\mathcal{H}$ takes a container data $c$ with a secret data $s$ to produce a modified container $\hat{c} = \mathcal{H}(c, s)$ that encodes $s$ while remaining visually similar to $c$. Then, a system $\mathcal{F}$ takes $\hat{c}$ to recover the secret data $s \sim \hat{s} = \mathcal{F}(\hat{c})$. In Audio-in-Image Steganography, $c$ is an image, and $s$ is an audio of variable length. Since the container's capacity is the limiting factor in what is capable of being hidden, we adopt $\gamma$ to represent the trade-off parameter between the reconstruction quality of the container and the secret data. Here, a larger value of $\gamma$ increases the quality of the audio reconstruction at the sacrifice of the container image quality, while a smaller value makes the change on the container image less noticeable at the sacrifice of the audio reconstruction quality.

The audio data goes through STFT (Short-Time Fourier Transform) during the preprocessing step and converts into the frequency domain. This makes $s, \hat{s} \in \mathbb{R}^{N \times T \times 2}$, with $N$ as the number of frequencies, $T$ as the varying length of the audio, and 2 as the real and complex channel. $c$ and $\hat{c}$ is

---

**Algorithm 1:** VLVQ Framework conditioned on $\gamma$

---

**Result:** Fully trained $\mathcal{H}$ and $\mathcal{F}$

Algorithm parameters: Container image $c$, Secret audio $s$, $\gamma$ range $\gamma_{min}, \gamma_{max}$, total iteration $T_{total}$, audio chunk number range $C_{min}, C_{max}$, frequency number and image resolution $h, w$;

Initialize $\mathcal{H}, \mathcal{F}$ with random weights;

**repeat** $T_{total}$ **times**

  Sample random integer $k$ in range $[C_{min}, C_{max}]$;
  Let $T = w \times k, c_1 = c$;
  Sample $\gamma$ from $\log(\gamma) \sim U(\log(\gamma_{min}), \log(\gamma_{max}))$;
  Sample $s \in \mathbb{R}^{h \times T \times 2}$ and $c \in \mathbb{R}^{h \times w \times 3}$ from the dataset;
  Crop $s$ into $s_i \in \mathbb{R}^{h \times w \times 2}, i = 1 \ldots k$;
  **for** $i = 1$ **to** $k$ **do**
    $c_{i+1} = \mathcal{H}(c_i, s_i, \gamma)$;
  **end**
  $\hat{c}_{k+1} = c_{k+1}$;
  **for** $i = k + 1$ **to** 2 **do**
    $\hat{c}_{i-1}, \hat{s}_{i-1} = \mathcal{F}(\hat{c}_i, \gamma)$;
  **end**
  $\mathcal{L}_{img} = \frac{1}{k} \sum_{i=2}^{k+1} ||c_i - c_1||_2$;
  $\mathcal{L}_{dec} = \frac{1}{k} \sum_{i=1}^{k} ||\hat{c}_i - \hat{c}_{i+1}||_2$;
  $\mathcal{L}_{stft} = \frac{1}{k} \sum_{i=1}^{k} ||s_i - \hat{s}_i||_2$;
  $\mathcal{L}_{total} = \mathcal{L}_{img} + \mathcal{L}_{dec} + \gamma \mathcal{L}_{stft}$;
  Backpropagate and optimize $\mathcal{H}$ and $\mathcal{F}$ with respect to $\mathcal{L}_{total}$;

**end**

---

an RGB image with $c, \hat{c} \in \mathbb{R}^{h \times w \times 3}$. We set $N = h = w$ for simplicity and only use $h$ and $w$ to denote spatial dimensions, thus $s, \hat{s} \in \mathbb{R}^{h \times T \times 2}$.

## 3.2 RECURSIVE PATCH INFERENCE

**Algorithm.** $T$ is a value that changes across different inputs. However, having $T = h = w$ would keep the spatial resolution of both $s$ and $c$ identical and constant, which is an ideal condition for inference on a deep neural network. We sample $s$ to have $T$ as a multiple of $w$ during training, so we can crop $s$ into $k = \frac{T}{w}$ chunks, with each chunk $s_i \in \mathbb{R}^{h \times w \times 2}, i = 1 \ldots k$. To guarantee this during inference, we pad $s$ to $s \in \mathbb{R}^{h \times w \lceil \frac{T}{w} \rceil \times 2}$ and crop $s$ into $k = \lceil \frac{T}{w} \rceil$ chunks.

**Encoding Process.** Let $c_i$ denote the container image with $i - 1$ audio chunks encoded inside. $c_1 = c$, and $c_{i+1} = \mathcal{H}(c_i, s_i)$, which denotes that the current container and a chunk of the secret data is fed into $\mathcal{H}$ to update the container, and it is recursively fed back into $\mathcal{H}$ with the next chunk of the secret data until $i = k + 1$. Finally, $\hat{c}_{k+1} = c_{k+1}$, which denotes the final container that conceals all the audio chunks.

**Decoding Process.** Then, $\hat{c}_{i-1}, \hat{s}_{i-1} = \mathcal{F}(\hat{c}_i)$, which recursively feeds the container image to $F$ to decode each audio chunks, while updating the container image to recursively feed itself back into $\mathcal{F}$. With this process, we obtain $\hat{s}_1 \ldots \hat{s}_k$ that corresponds to $s_1 \ldots s_k$. Figure 3 visualizes both the encoding and the decoding process.

## 3.3 OPTIMIZATION OBJECTIVE

We optimize both $\mathcal{H}$ and $\mathcal{F}$ with respect to $\mathcal{L}_{total}$, an objective function that linearly weighs three sub-objective functions.

**Audio Container Distance.** To ensure that the decoded audio is similar to the ground truth audio at the same chunk index, we minimize the average of their mean-squared distances $\mathcal{L}_{stft}$. Also, to minimize the change $\mathcal{H}$ makes on the container image, we minimize $\mathcal{L}_{img}$ which is the average of

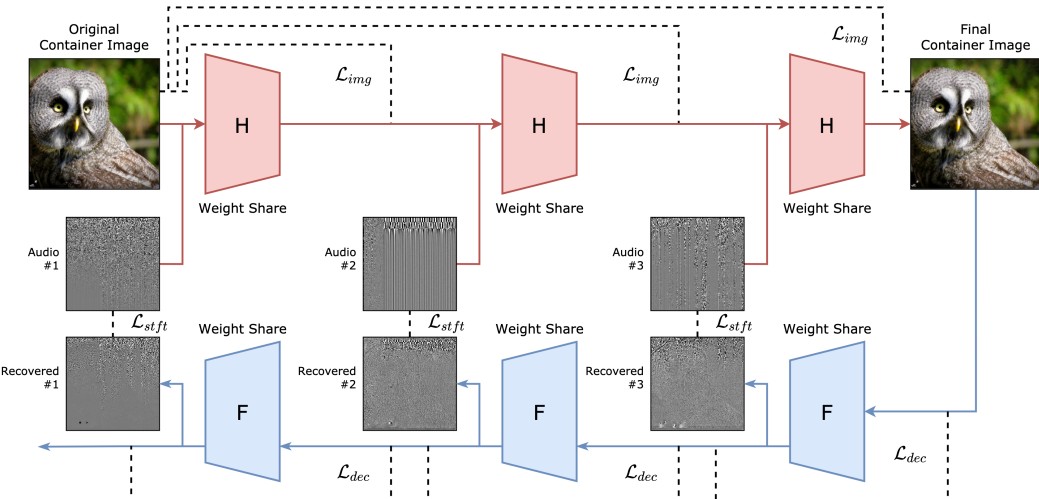

Figure 3: The overall process of the VLVQ framework. Iterative encoding on $\mathcal{H}$ minimizes the distance between original container image and the modified container image ($\mathcal{L}_{img}$). Iterative decoding on $\mathcal{F}$ minimally updates the container image ($\mathcal{L}_{dec}$), while making the reconstructed audio data to be as close to the ground truth ($\mathcal{L}_{stft}$).

the mean-squared distances between $c_1$ and $c_i$ for $i = 2 \dots k + 1$.

$$\mathcal{L}_{stft} = \frac{1}{k} \sum_{i=1}^{k} ||s_i - \hat{s}_i||_2 \quad \mathcal{L}_{img} = \frac{1}{k} \sum_{i=2}^{k+1} ||c_i - c_1||_2 \tag{1}$$

**Decoding Consistency.** $\mathcal{F}$ continuously updates the container image from the previous iteration. To prevent this process from destroying the information about the remaining audio chunks, it is essential to prevent $\mathcal{F}$ from making large modifications to the container. We achieve this by minimizing the mean-squared distance between the container images of two consecutive decoding iterations, $\mathcal{L}_{dec}$.

$$\mathcal{L}_{dec} = \frac{1}{k} \sum_{i=1}^{k} ||\hat{c}_i - \hat{c}_{i+1}||_2 \tag{2}$$

**Full Objective.** The final loss term $\mathcal{L}_{total}$ is a linear combination of the above three loss terms, and the hyperparameter $\gamma$ determines the trade-off between audio and image reconstruction quality. Figure 2 demonstrates the effects of this parameter by visualizing the changes of the residuals. We simply keep the coefficient of $\mathcal{L}_{dec}$ as 1.0, as it didn't have a significant effect on the outcome.

$$\mathcal{L}_{total} = \mathcal{L}_{img} + \mathcal{L}_{dec} + \gamma \mathcal{L}_{stft} \tag{3}$$

## 3.4 Trade-off Coefficient Conditioning

Modifying $\gamma$ in equation 3 gives control over the trade-off between audio reconstruction quality and container image quality. However, to allow such trade-offs at inference time with varying $\gamma$ terms, one must train several VLVQ frameworks, which would be computationally expensive. To address this, the VLVQ framework has the option to adopt a strategy of directly conditioning the models with $\gamma$, such that $\mathcal{H}(c_i, s_i)$ and $\mathcal{F}(\hat{c}_i)$ may become $\mathcal{H}(c_i, s_i, \gamma)$ and $\mathcal{F}(\hat{c}_i, \gamma)$.

**$\gamma$ Conditioning.** Both $\mathcal{H}$ and $\mathcal{F}$ are composed of several convolutional blocks, with each block following the order of convolution (Conv), normalization (Norm), and activation function (Act). Given the input and output of a block $\mathbf{X}, \mathbf{Y} \in \mathbb{R}^{H \times W \times C}$, with $\gamma \in \mathbb{R}^1, \gamma > 0$.

$$\mathbf{Y} = \text{Act}(\text{Norm}(\text{Conv}(\mathbf{X}))) \tag{4}$$

To condition $\gamma$, we adopt the FiLM conditioning layer of Perez et al. (2018), which does a linear transformation on the convolutional feature with scale and shift parameters generated from the conditional variable.

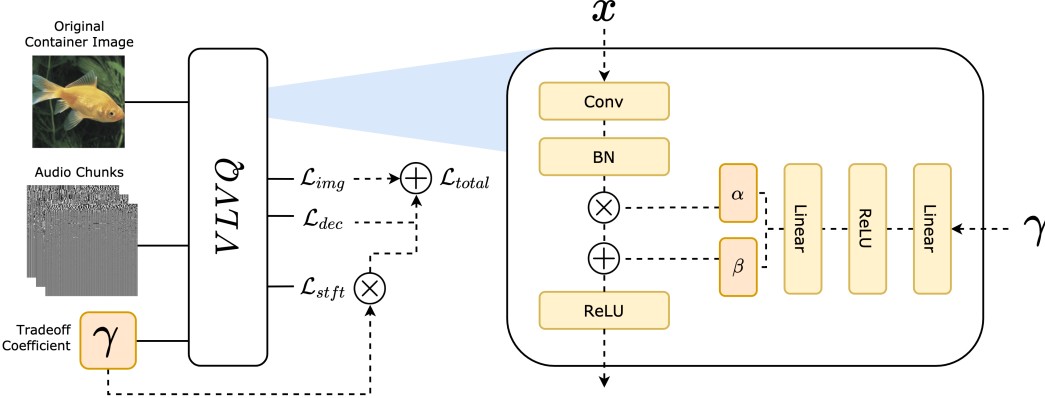

Figure 4: Conditioning module of the VLVQ framework. The architecture is conditioned on $\gamma$ via the FiLM layer (Perez et al., 2018), and the loss term is conditioned on $\gamma$ via linear combination.

$$\alpha = f(\gamma) \quad \beta = h(\gamma)$$
$$\mathbf{H} = \text{Norm}(\text{Conv}(\mathbf{X})) \quad (5)$$
$$\mathbf{Y} = \text{Act}(\alpha\mathbf{H} + \beta)$$

Here, $\alpha \in \mathbb{R}^C$, $\beta \in \mathbb{R}^C$, $H, Y \in \mathbb{R}^{H \times W \times C}$. $f$ and $h$ are two-layer fully-connected networks with ReLU activation. Figure 4 illustrates this conditioning module.

**Sampling Distribution of Gamma** Since $\gamma$ is no longer a hyperparameter, we sample $\gamma$ from a fixed distribution each training iteration. We apply a commonly used hyperparameter distribution of log-uniform distribution (Bergstra & Bengio, 2012) and find $\gamma_{min} = 0.01$ and $\gamma_{max} = 100.0$ to be a reasonable range.

$$\log(\gamma) \sim U(\log(\gamma_{min}), \log(\gamma_{max})) \quad (6)$$

## 4 EXPERIMENTS

Section 4.1 describe the implementation details of the VLVQ framework, and Section 4.2 describes the datasets and metrics used in our experiments. Section 4.3 demonstrates the effects of modifying the $\gamma$ parameter to both conditional and non-conditional versions of the framework, and Section 4.4 tests VLVQ's capability to extrapolate and encode variable length signals.

### 4.1 IMPLEMENTATION DETAILS

**Architecture.** We adopt a U-Net (Ronneberger et al., 2015) - like architecture for both $\mathcal{H}$ and $\mathcal{F}$. When conditioning the network on $\gamma$, we insert a FiLM layer (Perez et al., 2018) in all convolutional and transposed convolutional blocks. $\mathcal{H}$ receives a 5-channel input by concatenating $c_i$ and $s_i$, and $\mathcal{F}$ produces a 5-channel output which is split in the channel dimension into $\hat{c}_i$ and $\hat{s}_i$. Please refer to the supplementary materials for more details.

**Hyperparameters.** The network is trained end-to-end via Adam (Kingma & Ba, 2014) with the initial learning rate of 0.001, batch size of 4, and betas of 0.9 and 0.999 for two epochs. The learning rate decays by a factor of 10 halfway through the training. The framework is implemented on the PyTorch (Paszke et al., 2019) framework, and all models are trained on a single Nvidia Tesla V100 GPU.

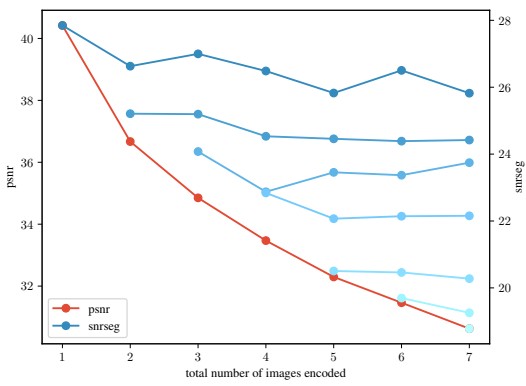

(a) Evaluating a non-condition model trained with $\gamma = 1.0$.

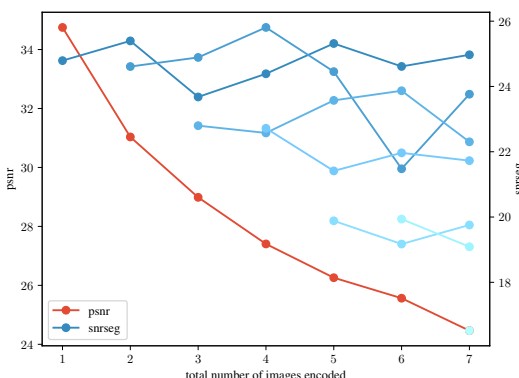

(b) Evaluating a condition model trained with $\gamma$ from $\log(\gamma) \sim U(\log(0.01), \log(100.0))$ and conditioned on $\gamma = 1.0$ during inference.

Figure 5: Evaluation of encoding variable number of chunks inside the VLVQ framework. Lighter color indicates the encoding of later audio chunks. Despite being trained on maximum three chunks, the model can be reasonably extrapolated to encode longer sequences.

## 4.2 EXPERIMENT SETUP

**Dataset.** For the images, we use the ILSVRC 2012 subset of the ImageNet (Russakovsky et al., 2015) dataset, containing 1.28 million natural images for training and 50k for validation. Images are preprocessed with a resize and a center crop of 224x224. For the audio, we use the AudioSet (Gemmeke et al., 2017) dataset, containing 1.7M audio clips that are each 10 seconds long. We convert all the samples into the frequency domain image with a height of 224 via STFT. After concatenating all the samples, one to three 224x224 random crops are taken from them every training iteration, with each crop representing an audio chunk that accounts for approximately 6 seconds of audio.

**Evaluation metrics.** For the container image quality, we measure PSNR (Peak Signal-to-Noise Ratio) between the original container image and the current container image (with the audio data embedded inside). For the audio reconstruction, we measure SegSNR (Segmental Signal-to-Noise Ratio) between the original audio and the reconstructed audio.

## 4.3 EFFECTS OF MODIFYING GAMMA

**Verifying Trade-offs.** Increasing the $\gamma$ parameter on VLVQ should increase the audio quality at the sacrifice of the container image quality and vice versa. Such trend is shown for both conditional and non-conditional models in Figure 6, demonstrating that modifications in $\gamma$ yield meaningful trade-offs between different objectives. An ideal conditional VLVQ framework should have near-identical results compared with the non-conditional VLVQ framework individually trained on their respective $\gamma$ parameters. However, a slight degradation is expected as the consequence of having a difficult optimization space that covers many values of $\gamma$, forcing the model to learn a distribution of solutions. Again, such trend is observed between the conditional and non-conditional model.

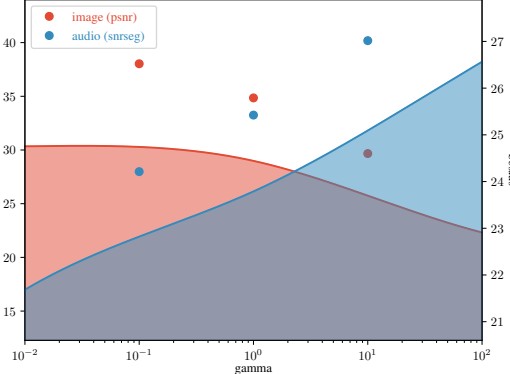

Figure 6: Conditional model (curve). Non-conditional models (point).

| Method | $\gamma$ | $psnr$ for different number of audio chunks | | | | | | |
|---|---|---|---|---|---|---|---|---|
| | | 1 | 2 | 3 | 4 | 5 | 6 | 7 |
| non-cond | 10.0 | 35.47 | 31.77 | 29.67 | 28.26 | 27.02 | 26.22 | 25.49 |
| | 1.0 | 40.42 | 36.67 | 34.85 | 33.47 | 32.30 | 31.46 | 30.63 |
| | 0.1 | 44.08 | 40.38 | 38.03 | 36.32 | 35.07 | 34.02 | 33.07 |
| cond | 10.0 | 31.40 | 27.85 | 25.75 | 24.37 | 23.04 | 22.64 | 21.78 |
| | 1.0 | 34.75 | 31.03 | 28.99 | 27.40 | 26.26 | 25.56 | 24.46 |
| | 0.1 | 36.06 | 32.13 | 30.29 | 28.48 | 26.99 | 26.29 | 25.02 |

| Method | $\gamma$ | $snrseg$ for different number of audio chunks | | | | | | |
|---|---|---|---|---|---|---|---|---|
| | | 1 | 2 | 3 | 4 | 5 | 6 | 7 |
| non-cond | 10.0 | 30.70 | 28.32 | 27.02 | 26.32 | 25.44 | 24.62 | 24.50 |
| | 1.0 | 27.85 | 25.92 | 25.42 | 24.18 | 23.26 | 22.76 | 22.07 |
| | 0.1 | 25.59 | 24.43 | 24.21 | 22.92 | 22.09 | 22.12 | 21.73 |
| cond | 10.0 | 25.11 | 27.73 | 25.09 | 24.67 | 23.49 | 23.50 | 22.94 |
| | 1.0 | 24.79 | 25.00 | 23.79 | 23.87 | 22.93 | 21.84 | 21.16 |
| | 0.1 | 25.12 | 22.16 | 22.82 | 20.65 | 19.84 | 19.63 | 18.49 |

Table 1: Effects of encoding different audio lengths (1 to 7 chunks) inside the VLVQ frameworks. The $snrseg$ values are averaged across the whole audio reconstruction. The effects of the trade-off becomes apparent as $\gamma$ decreases, with the increase in image quality and decrease in audio quality.

### 4.4 NUMBER OF ENCODED AUDIO CHUNKS

We test whether the VLVQ framework stands to the claim of being capable of encoding audio signals with varying length. To evaluate this, we test beyond the expected number of chunks during training ($C_{min} = 1, C_{max} = 3$) by evaluating up to 7 audio chunks. For both non-conditional and conditional framework, Figure 5a, 5b and Table 1 each shows that while both image and audio reconstruction quality drops with longer encoded sequences, the drop continues to maintain a gradual decline from 1 to 7 instead of a rapid drop at 4. This trend demonstrates VLVQ's reasonable extrapolation capability.

## 5 DISCUSSION

**Defenses Against Steganalysis.** While we make no claims regarding VLVQ's defensibility against steganalysis, future work may evaluate and equip VLVQ with defenses (Lyu & Farid, 2002; Fridrich, 2004; Fridrich & Kodovsky, 2012; Kodovsky et al., 2011; Qian et al., 2015) to prevent algorithmic detections.

**Failure Modes.** When dealing with multiple audio chunks, $\mathcal{F}$ sometimes fails to separate the signals between different chunks, resulting in the leakage of audio chunks into different timesteps. This usually happens when different audio chunks greatly vary in volume, as the louder chunks dominate the signals in the container image. Future work may employ an adversarial framework by introducing an adversarial framework to discourage $\mathcal{F}$ from producing signals that can be classified as originating from a different chunk.

**Conditioning Methods** While we used FiLM on all convolutional blocks due to its simple architecture choice, one may search for a better method of conditioning $\gamma$, such as searching for the optimal layers to place the conditioning module.

**Limitations.** Although a simple U-Net (Ronneberger et al., 2015) was sufficient to yield satisfactory results, there are clear limitations to this architecture. For example, a convolutional layer's inductive bias forces the network to rely mostly on local information. Figure 2 demonstrates that most residuals are visible at the top of the image, where the most prominent STFT features lie. To exploit the full capacity of the container image, architectures with minimal inductive biases such as transformers (Vaswani et al., 2017) may be employed.

## 6 CONCLUSION

In this work we present VLVQ, a steganographic framework that encodes variable length audio data into images with varying quality trade-offs. Compared to other frameworks of this kind, VLVQ enables a greater degree of freedom in terms of secret audio length and infernce-time quality trade-offs. We achieve this through recursive inference of multiple audio chunks and conditioning the model with the trade-off parameter via FiLM layers. Experiments on ImageNet and Audioset verify these claims and demonstrate extrapolation capabilities beyond audio length seen during training. We hope these efforts make audio-in-image steganography feasible in diverse real-world scenarios.

## A ETHICS STATEMENT

With growing concerns over various societal issues regarding individual privacy, tools such as steganography can empower individuals to have greater control over their information. For example, activists in nations where encryption is criminalized may apply steganography in a non-secret medium to hide their messages. In this aspect, our work helps widen the available pool of steganographic medium an individual can hide their information on. But this raises concern to a potential risk based on the malicious uses of steganography, such as its applications as digital fingerprints in intelligence services. Hence, researches on steganography and its applications must be developed with these positive and negative effects in consideration.

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
