# OpenReview forum: "Variable Length Variable Quality Audio Steganography"
_ICLR.cc/2022/Conference — ICLR 2022 Submitted_

### Official Review · Reviewer_c5tq · 2021-10-21

**Correctness:** 3
**Technical Novelty And Significance:** 3
**Empirical Novelty And Significance:** 2
**Recommendation:** 3
**Confidence:** 5

**Main Review:**

The audio specific aspect of the problem is not sufficiently well justified. Generally speaking, in steganography, the nature of the hidden message does not matter : it can be audio, image or video signals and so long, the only assumption is that the message is binary which is always possible when dealing with digital messages. So why embedding a message that is an audio should lead to any specific difficulties or conversely why would it be helpful to design better steganography ? This should be highlighted in the introduction of the paper.

The variable length message aspect of the problem is partially legitimate : several steganography algorithms will need to be re-trained or re-adjusted when the size of the message / payload changes. Some even fail to generalize to any other payload than the one they saw during training. For the specific case of ML-based steganography, this raises a well known problem of OOD generalization. But it must also be underlined that many steganographic algorithms can embed messages with various lengths and require no particular effort to do so. This means that the proposed approach should be compared to such methods in the experimental section of the paper. As a baseline, the approach could be compared to UNIWARD [1] coupled with STC encoding [2].

[1] Holub, V., Fridrich, J., & Denemark, T. (2014). Universal distortion function for steganography in an arbitrary domain. EURASIP Journal on Information Security, 2014(1), 1-13.

[2] Filler, T., Judas, J., & Fridrich, J. (2011). Minimizing additive distortion in steganography using syndrome-trellis codes. IEEE Transactions on Information Forensics and Security, 6(3), 920-935.

Obviously, the authors address a problem that is different from the one addressed by the mainstream steganographic literature. They allow themselves to modify message length by degrading the audio signal. This is where using an audio is instrumental from the authors’ standpoint : reaching a compromise between the audio signal quality and embedding capacity of the cover image.
The legitimacy of this very problem is itself debatable. For a genuinely secure steganography, one would (i) lower the quality of the signal to the minimal acceptable value so that the message is as small as possible and (ii) chose an embedding that minimizes detectability for this message size. Indeed, this strategy minimizes the overall detectability which prevails over audio quality.

Moreover, the proposed approach raises two major security issues :
	•	if one wishes to encrypt the secret message prior to insertion in the cover image, then failing to perfectly extract the message from the stego image will make it impossible to decipher and thus useless.
	•	there is no notion of secret key in the authors’ approach. Consequently, an attacker can train a neural net similarly and find hidden messages pretty easily.


Another problematic issue is that the notion of detectability is understood as the accuracy of SOTA steganalysis deep neural nets such as SRnet [3] or XUnet [4]. The authors did not evaluate their approach wrt these opponents and thus fail to prove the operational interest of their steganographic scheme.

[3] Boroumand, M., Chen, M., & Fridrich, J. (2018). Deep residual network for steganalysis of digital images. IEEE Transactions on Information Forensics and Security, 14(5), 1181-1193.

[4] Xu, G. (2017, June). Deep convolutional neural network to detect J-UNIWARD. In Proceedings of the 5th ACM Workshop on Information Hiding and Multimedia Security (pp. 67-73).

Finally, using SFTF creates a redundant representation of the initial audio signal. So the message length is inflated by this process which is again is not good for undetectability.

**Summary Of The Paper:**

This submission deals with steganography, i.e. the task of hiding a secret message inside a multimedia content. The authors in this submission choose images as cover content and audio signals as secret messages. They also ask their embedding algorithm to be flexible in the sense that variable length audios can be processed by the same algorithm.
This is achieved by cleverly designing a neural network pipeline with recursions to tackle the variable length message issue.



**Summary Of The Review:**

Pros :
- the authors circumvent the variable message length difficulty by recursively embedding message chunks using a neural network.
- the weight regulating quality/detectability trade-off is tunable at test time thanks to loss-conditional training

Cons :
- the authors evaluate message undetectability through SNR measures instead of relying on detectability by a SOTA opponent. The proposed insertion is likely to be easily detectable by such an opponent because it is not constrained to preserve natural image statistics.
- the problem addressed is steganography without a shared secret key and thus relies on security through obscurity which was rejected as a sufficiently secure framework long ago. I believe the authors should look for other use cases. The approach might be seen as a form of uncanny compression : instead of sending an image + an audio, one can send an image containing the audio.

---

### Official Review · Reviewer_gihE · 2021-11-02

**Correctness:** 2
**Technical Novelty And Significance:** 2
**Empirical Novelty And Significance:** 2
**Recommendation:** 5
**Confidence:** 2

**Details Of Ethics Concerns:**

Not applicable for this work.

**Main Review:**

Strengths:
1) The paper is well organised and moreover, the research pursued is very important in the cyber security field.
2) In section 4 (Experimental section), the varying trade-offs of gamma parameter and data on varying length of signal hiding could be valuable/important for researchers in the field of steganography.

Weaknesses:
The paper can be improved based on the following points:
1) In section 1, the authors talk about the drawback of variable length hiding, however, the definition of variable length is not mentioned. Please mention this as it could be any length.
2) In section 2, several of the recent works in steganography/deep learning have been missed and would be great to mention them as well.
3) In section 3 (method), it would be better to connect the algorithm (Algorithm 1) mentioned in the text body such that readers find it easier to follow.
4) The encoding and decoding process lacks novelty. It would be better to justify why such a method has been chosen, apart from the reason of enabling variable length signal hiding. The point of the paper is to contribute to the scientific knowledge of the community, however, the proposed method felt more as applicative based method where a new design proposal is made rather than the advancement of the scientific field. The design justification is missing from the Method's section.
5)  In section 3.3 and 3.4, the symbols used in the equations are also not described/defined intuitively, which could be a hindrance for many readers. Please update that so that readers of broader interest find it interesting to understand and read the proposed work as well.
6) In section 4, the experiments are lacking in comparative study. There has been some methods in the same domain to utilize steganography/deep learning to hide signal, however, the proposed method has not been compared with them to show the efficacy of the proposed. The authors should consider this.


**Summary Of The Paper:**

In the paper, the authors propose a method to  hide audio inside an image with variable length and variable quality leveraging a deep learning based steganographic method framework capable of hiding variable-length audio inside an image by
training the network to iteratively encode and decode the audio data from the container image.


**Summary Of The Review:**

The paper can be greatly improved based on the points mentioned in the weaknesses section in the main review for the authors. Please refer to the points before the paper could be considered for presentation at ICLR.

---

> ### Comment · Reviewer_gihE · 2021-11-25
> **Reply to updated review**
>
> Thank you for all the different viewpoints mentioned in the review, however, after going through all the comments/review feedback I have noticed that all the reviewers recommended negative against the paper. So, my final decision remains as "marginally below acceptance threshold".

---

### Official Review · Reviewer_zNiL · 2021-11-07

**Correctness:** 3
**Technical Novelty And Significance:** 2
**Empirical Novelty And Significance:** 2
**Recommendation:** 5
**Confidence:** 3

**Main Review:**

Although the idea and techniques are not surprising to me, the proposed method seems to have responded to a very common situation (variable audio length, flexible quality-unnoticeability trade-off) in real-world applications. The presentation is also satisfying. Nevertheless, to my understanding, the paper does not seem to have implemented the idea in a comprehensive and compelling way.

Method:
* I hope the authors could explain more on the well-definedness of the task. In Sec. 3.1, the task is to learn the inverse of the map $\hat{c} = H(c,s)$, but how is this $H$ invertible? Its output has more dimensions than its input. One possibility may be that images have a certain degree of freedom to be perceptually the same, and this degree of freedom can be leveraged for encoding an audio signal. If it is the case, is the capacity of the degree of freedom large enough to hold an audio signal? I expect such an explanation to convince me the task is well-defined, and to provide an insight of the capacity to encode an audio signal.
* Is the audio encoding of equal quality if encoded in a different order? More specifically, does the audio reconstruction $\Vert s - F(H(c_i, s)) \Vert$ change with $i$? If it does, the order of the chunks matters, and the decoded audio may have decreasing quality.
* Will the image become more and more distorted as the length of the audio increases?

Performance:
* In Fig. 1, the decoded audio signal still seems to hold the pattern of the image, e.g. the bird's head and the owl's head. How does it sounds like? Also the audio-encoded images still have a sensible flaw, particularly on the top region.
* Fig. 2 seems not mentioned in the text. Also, are the second to fourth images showing audio-encoded images or the difference from the original image? In either case, even with $\gamma=0.1$ the modification to the image seems quite noticeable.
* The experiment does not seem to compare with baseline methods. I'm not expert in such steganography so I'm not sure if there is indeed no variable-length steganography method, but at least I expect a comparison with a baseline method in a fixed-length setup.


**Summary Of The Paper:**

The paper presents a method for steganography, that is to encode and decode an audio signal into an image in an unnoticeable way. The feature of the proposed method is that it can handle audio with variable length, and can flexibly trade-off between audio encoding quality and image modification unnoticeability in the inference stage. The method is to learn a pair of encoder and decoder to achieve both encoding and unnoticeability up to some trade-off, and the trade-off coefficient is also input into the model to avoid retraining for an arbitrary trade-off in inference.

**Summary Of The Review:**

The paper contributed a method that seems to handle a common case in practice. But the technique does not seem novel enough, the discussion on the method does not seem necessarily comprehensive, and the performance does not seem convincing.

---

### Official Review · Reviewer_jZgM · 2021-11-07

**Correctness:** 4
**Technical Novelty And Significance:** 3
**Empirical Novelty And Significance:** 3
**Recommendation:** 5
**Confidence:** 4

**Main Review:**


Strengths

The paper is written well and the approach is explained clearly. Iteratively encoding and decoding the audio data in the image is a strength. An optional conditional loss term is also a plus.

Weakness

While the approach is promising, the paper still needs some more work. Some of those points are mentioned in the Discussion section.

First, it is difficult to assess how the method will perform if post-processing operations such as JPEG compression, smoothening, blurring and other operations are performed on the image.

It is also difficult to evaluate the proposed approach without knowing how a stegnanlysis method would work on the proposed method.

It is also not clear how many seconds of variable length audio are considered in the experiments. Though some details are presented in Table 1, the exact lengths of the audio in seconds is not clear.


**Summary Of The Paper:**

This paper presents a steganographic approach called Variable Length Variable Quality Audio Steganography (VLVQ) that encodes variable length audio data inside images with varying quality trade-offs. The method works by training a network to iteratively encode and decode the audio data from the container image. An optional conditional loss term is also proposed apart from the standard reconstruction loss. Experiments presented on two large datasets - ImageNet dataset for container images and AudioNet dataset for embedding the audio, shows that the approach is promising.



**Summary Of The Review:**

Though the paper has some merit and the approach of iteratively embedding audio in the encoder-decoder is interesting, the paper needs some more work, which if presented will make the paper stronger.

---

### Decision · Program_Chairs · 2022-01-20

**Decision:**

Reject

**Comment:**

This paper presents a steganographic approach called Variable Length Variable Quality Audio Steganography (VLVQ) that encodes variable length audio data inside images with varying quality trade-offs. However, according to the reviewers, the proposal made in this paper is not novel enough, there are many details missing in the paper, and the experimental study is far from comprehensive and conclusive. Afte the reviewers provided their comments, the authors did not submit their rebuttals. Therefore, as a result, we do not think the paper is ready for publication at ICLR.